 **eLIFE**

# Akt-mTORC1 signaling regulates Acly to integrate metabolic input to control of macrophage activation

Anthony J Covarrubias[1], Halil Ibrahim Aksoylar[1], Jiujiu Yu[1], Nathaniel W Snyder[2,3], Andrew J Worth[2], Shankar S Iyer[4], Jiawei Wang[5], Issam Ben-Sahra[1], Vanessa Byles[1], Tiffany Polynne-Stapornkul[1], Erika C Espinosa[1], Dudley Lamming[6], Brendan D Manning[1], Yijing Zhang[5], Ian A Blair[2], Tiffany Horng[1]*

[1]Department of Genetics and Complex Diseases, Harvard T.H. Chan School of Public Health, Boston, United States; [2]Center of Excellence in Environmental Toxicology, University of Pennsylvania, Philadelphia, United States; [3]A.J. Drexel Autism Institute, Drexel University, Philadelphia, United States; [4]Department of Medicine, Brigham and Women's Hospital, Boston, United States; [5]Institute for Plant Physiology and Ecology, Shanghai Institute for Biological Sciences, Chinese Academy of Sciences, Shanghai, China; [6]Department of Medicine, University of Wisconsin-Madison, Madison, United States

*For correspondence: thorng@hsph.harvard.edu

**Competing interests:** The authors declare that no competing interests exist.

**Abstract** Macrophage activation/polarization to distinct functional states is critically supported by metabolic shifts. How polarizing signals coordinate metabolic and functional reprogramming, and the potential implications for control of macrophage activation, remains poorly understood. Here we show that IL-4 signaling co-opts the Akt-mTORC1 pathway to regulate Acly, a key enzyme in Ac-CoA synthesis, leading to increased histone acetylation and M2 gene induction. Only a subset of M2 genes is controlled in this way, including those regulating cellular proliferation and chemokine production. Moreover, metabolic signals impinge on the Akt-mTORC1 axis for such control of M2 activation. We propose that Akt-mTORC1 signaling calibrates metabolic state to energetically demanding aspects of M2 activation, which may define a new role for metabolism in supporting macrophage activation.

## Introduction

Macrophages are pleiotropic cells that assume a variety of functions depending on tissue of residence and tissue state. Their ability to acquire diverse, context-dependent activities requires activation (or polarization) to distinct functional states, triggered by various factors including microbial products, cytokines, and growth factors (*Davies et al., 2013*; *Murray and Wynn, 2011*). M1 or classical activation is triggered during infection by microbial products including LPS, leading to the transcriptional upregulation of genes encoding antimicrobial activities and inflammatory cytokines. M2 or alternative activation is triggered by IL-4 and IL-13 produced during parasite infections, and activates the transcription factor Stat6 to induce a transcriptional program that coordinates fibrosis, tissue remodeling, and Type 2 inflammation (*Davies et al., 2013*; *Murray and Wynn, 2011*). Therefore, the induction of multi-component transcriptional programs underpins macrophage activation.

While macrophage activation is relatively well-understood at the level of signal transduction, transcriptional regulation, and acquisition of new effector activities, the metabolic underpinnings remain less clear. An emerging view is that macrophage activation to particular states is associated with

**eLife digest** Macrophages are immune cells that are found in most of the tissues of the body. Exactly what the macrophages do depends on which tissue they are in, and the state of the tissue. For example, M2 macrophages can multiply in numbers, heal wounds or help to fight off parasites depending on the signals they receive from their environment. Conversely, when macrophages sense pathogens such as bacteria they can also become M1 macrophages, which produce inflammatory molecules that help kill the invading bacteria.

As a macrophage transforms into a more specialized state, its metabolism – the set of chemical reactions the cell performs in order to survive and thrive – also changes. This shift appears to play an important role in activating the macrophages and determining how they'll specialize. However, little is known about how metabolism exerts this control.

The metabolism of a cell can be investigated in part by studying the molecules, or "metabolites", that the cell produces. Covarrubias et al. studied what happens when unspecialized macrophages from mice were activated by a signaling molecule called IL-4. This signaling molecule causes the cells to become M2 macrophages, and the experiments revealed that IL-4 signaling controls the amount of a metabolite called acetyl-CoA in the cells.

Acetyl-CoA can influence how the DNA of a gene is packaged in a cell, and thus affect whether a gene is switched on and "expressed" or not. Covarrubias et al. therefore also analyzed a major metabolic sensing pathway – the Akt-mTORC1 pathway – and showed how this pathway was able to act as a nutrient sensor for the macrophage and control the enzyme responsible for making acetyl-CoA. Therefore, the Akt-mTORC1 pathway can control the level of gene expression changes in the macrophages as a result of IL-4 signaling.

The analysis showed that the increase in acetyl-CoA levels increases the expression of some of the genes that cause the M2 macrophages to change state and develop their specialist behaviors. However, only a subset of these genes – those that encode metabolically demanding activities such as immune cell trafficking – have their expression controlled in this way. Further studies are now needed to investigate whether other macrophage types use the same pathways to control their responses.

distinct metabolic shifts (*Pearce and Everts, 2015*; *Galván-Peña and O'Neill, 2014*; *Biswas and Mantovani, 2012*). For example, M1 macrophages upregulate glucose and glutamine utilization (*Tannahill et al., 2013*; *Cramer et al., 2003*), while M2 macrophages augment β-oxidation and glutamine consumption (*Vats et al., 2006*; *Jha et al., 2015*). Importantly, such metabolic shifts critically support macrophage activation. Increased glycolytic flux in M1 macrophages is coupled to de novo lipogenesis, which enables ER and Golgi expansion and production of high levels of inflammatory cytokines (*Everts et al., 2014*). Another consequence of enhanced glycolysis is accumulation of the TCA cycle metabolite succinate, leading to stabilization of the transcription factor HIF-1α and transcriptional induction of *Il1b* and other target genes in the M1 macrophage (*Tannahill et al., 2013*). How oxidative metabolism boosts M2 activation is not clear, but glutamine metabolism fuels production of UDP-GlcNAC, an important modification of multiple M2 markers (*Jha et al., 2015*).

Consistent with the idea that macrophage activation is supported by metabolic shifts, recent studies indicate that macrophage polarizing signals impinge on metabolic signaling pathways. Polarizing signals like LPS and IL-4 regulate the activity of Akt, mTORC1, and AMPK (*Everts et al., 2014*; *Byles et al., 2013*; *Cheng et al., 2014*; *Weichhart et al., 2008*), presumably to coordinate metabolic processes that critically underlie macrophage polarization. Limited studies indicate that perturbing the activity of these metabolic regulators impairs macrophage metabolism and activation (*Everts et al., 2014*; *Cheng et al., 2014*). For example, Akt mediates enhanced glycolysis to support lipid synthesis and inflammatory cytokine secretion in M1 macrophages (*Everts et al., 2014*). Akt similarly stimulates glucose-fueled lipid synthesis in growing and proliferating cells, where lipids are used to build cellular membranes (*Robey and Hay, 2009*). Therefore, M1 macrophages co-opt a metabolic process (Akt-dependent lipogenesis) in order to coordinate a macrophage-specific function

(inflammatory cytokine secretion). In general, however, how polarizing signals control metabolic shifts, and the full implications of this for control of macrophage activation, remains poorly understood.

Here we show that integration of the Akt-mTORC1 pathway into IL-4 signaling allows for selective control of some M2 responses. Control is exerted at the level of Acly, a key enzyme in Ac-CoA production, thereby modulating histone acetylation and transcriptional induction of a subset of M2 genes. Consistent with its role as an important metabolic sensor, the Akt-mTORC1 pathway couples metabolic input to such gene-specific control. Our findings also reveal subsets of the M2 response, including chemokine production and cellular proliferation, that are linked to metabolic state by Akt-mTORC1 signaling.

## Results

### Akt regulates increased glucose metabolism in M2 macrophages

Akt is a major metabolic regulator implicated in M2 activation (*Byles et al., 2013*; *Ruckerl et al., 2012*), but the underlying mechanisms remain poorly characterized. To begin to address this question, we employed unbiased metabolic profiling of M2 macrophages, using LC/MS-based metabolomics and a platform that measures ~290 small metabolites representative of all major pathways of intermediary metabolism (*Ben-Sahra et al., 2013*). Top enriched pathways include urea cycle and arginine and proline metabolism, consistent with previous studies indicating upregulation of arginine metabolism in M2 macrophages (*Van Dyken and Locksley, 2013*), as well as amino acid utilization and metabolism and nucleotide metabolism (*Figure 1A*, *Supplementary file 1*). Other top enriched pathways include glycolysis, amino sugar metabolism, and glycine, serine, and threonine metabolism, suggesting altered flux through glycolysis and glycolytic shunts (*Figure 1A*, *Supplementary file 1*).

As M2 activation is thought to be sustained by fatty acid rather than glucose utilization (*Cramer et al., 2003*; *Vats et al., 2006*), we decided to re-examine the role of glycolysis in M2 macrophages. We found that BMDMs increased glucose uptake in a time-dependent manner in response to IL-4 treatment. Such increase was reduced by cotreatment with the Akt inhibitor MK2206 (*Figure 1B*), indicating control by Akt and consistent with a role for Akt in regulating glycolysis in many settings (*Robey and Hay, 2009*). Moreover, enhanced glucose consumption in M2 macrophages was associated with an Akt-dependent increase in both glycolysis and oxidative metabolism, as indicated by extracellular flux assays (*Figure 1C*). Importantly, glycolytic flux was needed for optimal implementation of the M2 program. Similar to the β-oxidation inhibitor etomoxir, the glycolysis inhibitor 2-DG reduced IL-4-mediated induction of some M2 genes (*Figure 1D*). Therefore, Akt mediates enhanced glucose consumption in M2 macrophages, and this contributes to induction of M2 gene expression. Such glucose consumption may also fuel production of UDP-Glc-NAc, the substrate for glycosylation of some M2 markers (*Jha et al., 2015*). In contrast, Akt does not control β-oxidation in M2 macrophages (*Figure 1E*).

### IL-4 signaling activates Akt to allow for selective control of M2 gene induction

Because the increase in glucose utilization was relatively modest, we considered that Akt could play additional roles in control of M2 activation and turned to an analysis of M2 gene regulation. We examined induction of *Retnla, Arg1, Mgl2, Chi3l3, Cd36*, and *Fabp4*, "hallmark" M2 genes commonly used in studies of M2 activation (*Van Dyken and Locksley, 2013*). Consistent with the role of Stat6 as a transcriptional master regulator of M2 activation (*Odegaard and Chawla, 2011*), induction of these M2 genes was ablated in Stat6 KO BMDMs (*Figure 2—figure supplement 1A*). Importantly and as reported (*Byles et al., 2013*; *Ruckerl et al., 2012*), Akt activity controlled the induction of a subset of M2 genes. In the presence of the Akt inhibitor MK2206, induction of *Arg1, Retnla*, and *Mgl2* was reduced ~40–80%, while *Chi3l3, Cd36*, and *Fabp4* were not affected (or even superinducible) (*Figure 2A*). Use of a structurally distinct Akt inhibitor, Aktviii, yielded similar results, suggesting specificity in inhibition (data not shown). Below, these two groups of genes will be referred to as Akt-dependent and Akt-independent M2 genes, respectively.

The IL-4R activates Jak-Stat signaling as well as Akt-mTORC1 signaling in macrophages (*Byles et al., 2013*) (*Figure 2—figure supplement 1B*). Receptor ligation activates the latent activity of Jak1 and Jak3 kinases, leading to phosphorylation and activation of Stat6, as well as engagement

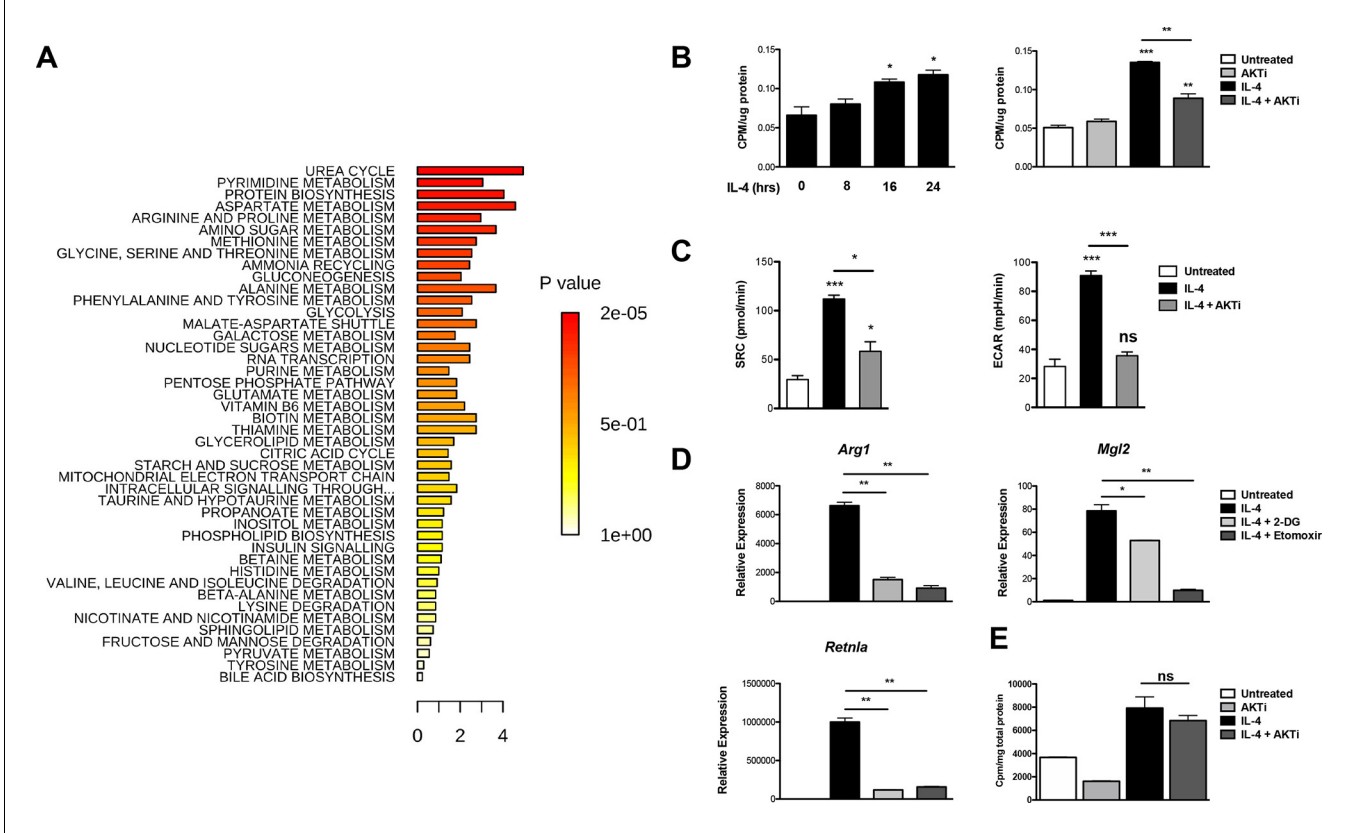

**Figure 1.** Akt regulates enhanced glucose utilization in M2 macrophages. (**A**) Top metabolic pathways enriched in macrophages stimulated for 12 hr with IL-4 (relative to unstimulated macrophages) as identified by LC/MS-based metabolomics profiling. (**B**) M2 macrophages increase glucose uptake in an Akt-dependent manner. BMDMs were treated with IL-4 for the indicated time periods (left) or 16 hr +/- the Akt inhibitor MK2206 (Akti) (right), followed by analysis of uptake of $^3$H-deoxy-D-glucose. (**C**) Increased glucose utilization in M2 macrophages is associated with enhanced oxidative metabolism and glycolysis. BMDMs were treated with IL-4 for 20 hr +/- Akt inhibitor, followed by analysis of spare respiratory capacity (SRC) and aerobic glycolysis (ECAR) in extracellular flux analyses. (**D**) M2 gene induction is sensitive to the glycolysis inhibitor 2-deoxyglucose (2-DG). BMDMs were treated with IL-4 for 16 hr +/- 2-DG or the β-oxidation inhibitor etomoxir pretreatment, followed by analysis of M2 gene induction by qPCR. (**E**) Akt does not regulate β-oxidation in M2 macrophages. BMDMs stimulated for 36 hr with IL-4 +/- Akt inhibitor pretreatment were incubated for 3 hr with $^3$H-palmitate for analysis of β-oxidation. The student's t-test was used to determine statistical significance, defined as *P<0.05, **P<0.01, and ***P<0.001.

of the adaptor protein IRS2. IRS2 recruits PI3K, which generates PIP3 from PIP2 leading to phosphorylation and activation of Akt. Activated Akt phosphorylates and inactivates the TSC complex, a negative regulator of mTORC1, to activate mTORC1. While the precise relationship between Jak-Stat and Akt-mTORC1 signaling remains unclear, the data in *Figure 2A* and *Figure 2—figure supplement 1A* suggest that they may operate in parallel and independently downstream of the IL-4R. Indeed, IL-4-mediated increases in Stat6 activation, as indicated by phosphorylation on Y641, was not affected in the presence of an Akt inhibitor (*Figure 2B*). Stat6 activity as measured by a Stat6-dependent luciferase reporter was also not impaired by inhibition of Akt activity (*Figure 2—figure supplement 1C*). Conversely, WT and Stat6 KO BMDMs could similarly activate Akt, as indicated by phosphorylation on S473, as well as mTORC1, as indicated by phosphorylation of the mTORC1 target S6K, in response to IL-4 (*Figure 2B*). These findings support the idea that the Jak-Stat and Akt-mTORC1 pathways are independent signaling branches downstream of the IL-4R, and suggest a basis by which all M2 genes are controlled by Stat6 while a subset receives additional inputs from the Akt-mTORC1 pathway.

How might Akt signaling regulate a subset of M2 genes? A seminal study from Wellen and colleagues indicated that in cancer cells and differentiating adipocytes, metabolic state is linked to gene expression via effects on histone acetylation (*Wellen et al., 2009*), thus we hypothesized that Akt may

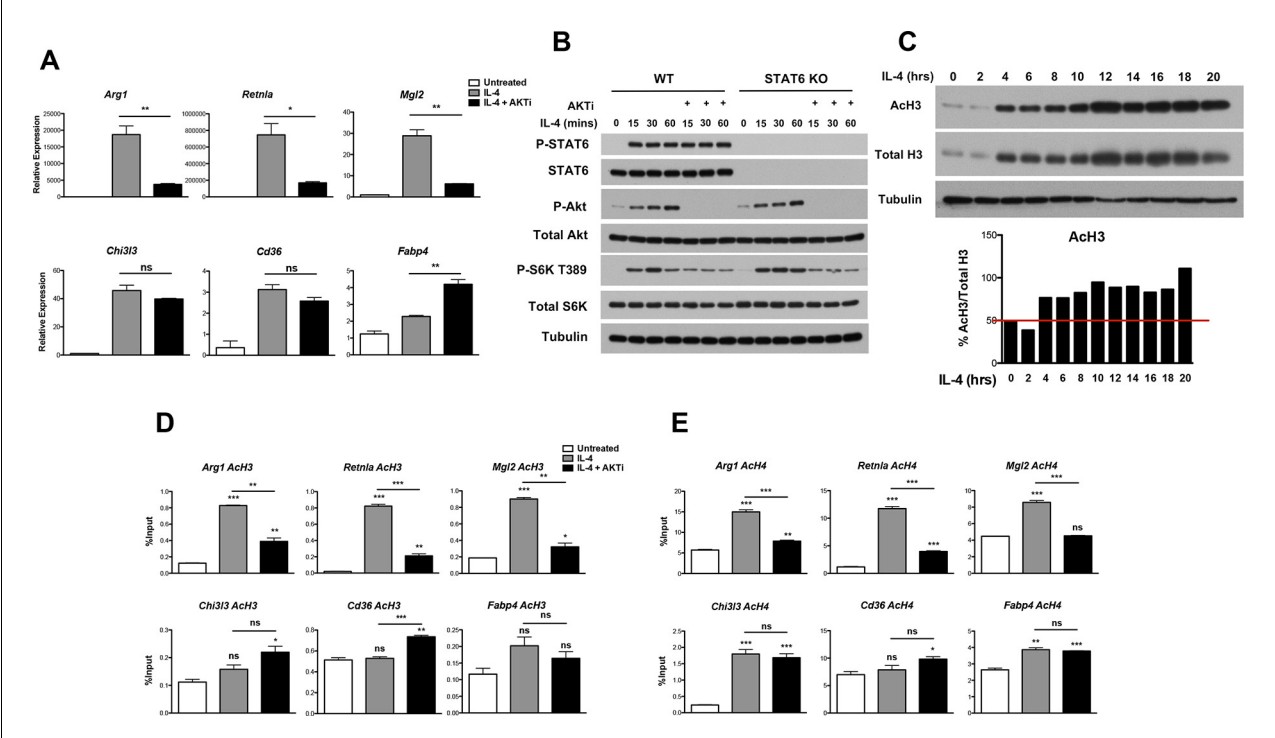

**Figure 2.** Akt regulates inducible histone acetylation at some M2 genes. (**A**) Akt activity stimulates induction of a subset of M2 genes. BMDMs were stimulated with IL-4 for 16 hr +/- the Akt inhibitor MK2206 (Akti) pretreatment, followed by analysis of M2 gene induction by qPCR. (**B**) The Jak-Stat and Akt-mTORC1 pathways are activated independently downstream of the IL-4R. WT and Stat6 KO BMDMs were stimulated with IL-4 +/- Akt inhibitor as indicated. Analysis of Stat6, Akt, and mTORC1 activation was assessed by western blotting. (**C**) IL-4 induces a global increase in histone H3 acetylation. BMDMs were stimulated with IL-4 over the time course indicated, followed by analysis of histone H3 acetylation by western blotting. *Bottom*, quantitation of acetylated H3 over total H3. (**D, E**) Akt regulates inducible H3 (**D**) and H4 (**E**) acetylation at some M2 genes. BMDMs stimulated with IL-4 for 16 hr +/- Akt inhibitor pretreatment were subject to ChIP analysis using antibodies to acetylated H3 or acetylated H4. Enrichment of the indicated M2 gene promoters was assessed by qRT-PCR. The student's t-test was used to determine statistical significance, defined as *$P<0.05$, **$P<0.01$, and ***$P<0.001$.

The following figure supplements are available for figure 2:

**Figure supplement 1.** Stat6 and Akt-mTORC1 pathways are independent signaling branches downstream of the IL-4R.

**Figure supplement 2.** Akt regulates inducible histone acetylation at some M2 genes.

control histone acetylation to regulate M2 gene expression. Indeed, IL-4-treatment of BMDMs enhanced global acetylation of H3 and H4 histones, as indicated by western blot of whole cell lysates (*Figure 2C*, *Figure 2—figure supplement 2A,B*). Importantly, IL-4-inducible increases in global H3 and H4 acetylation were reduced by cotreatment with an Akt inhibitor, indicating at least partial dependence on Akt (*Figure 2—figure supplement 2A,B*). In contrast, tubulin acetylation was not modulated by IL-4 treatment (*Figure 2—figure supplement 2A,B*). We next examined gene-specific patterns of H3 and H4 acetylation by chromatin immunoprecipitation (ChIP) experiments. IL-4 treatment increased H3 and H4 acetylation at promoters of M2 genes (*Figure 2D,E*, *Figure 2—figure supplement 2C*), with the degree of inducible acetylation correlating fairly well with the degree of gene induction (*Figure 2A*). Interestingly, such increases in H3 and H4 acetylation were reduced by an Akt inhibitor at M2 genes induced in an Akt-dependent manner (*Arg1, Retnla, Mgl2*), but not at M2 genes induced independently of Akt (*Chi3l3, Cd36, Fabp4*) (*Figure 2D,E*). Pol II recruitment to M2 gene promoters paralleled H3 and H4 acetylation, and was controlled by Akt at M2 genes induced in an Akt-dependent manner (*Figure 2—figure supplement 2D*). Together, these findings support the hypothesis that Akt regulates histone acetylation and Pol II recruitment at a subset of M2 genes.

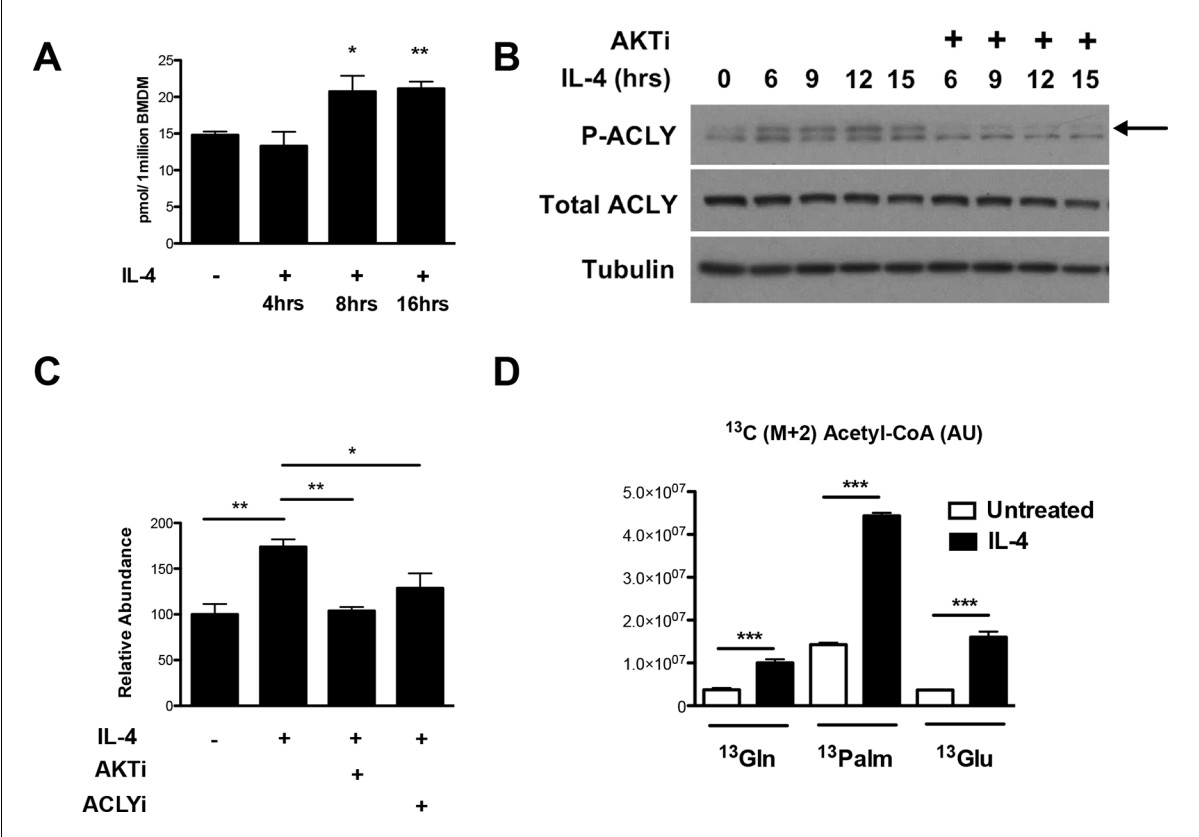

**Figure 3.** The Akt-Acly axis regulates inducible Ac-CoA production in M2 macrophages. (A) IL-4 treatment increases Ac-CoA production. BMDMs were stimulated for the indicated time periods with IL-4, followed by analysis of Ac-CoA levels by LC-MS. (B) Akt regulates IL-4-inducible Acly phosphorylation. BMDMs were stimulated as indicated, followed by analysis of Acly phosphorylation by western blotting. Arrow indicates phospho-Acly. (C) Akt and Acly regulate IL-4-inducible production of Ac-CoA. BMDMs stimulated for 16 hr with IL-4 +/- inhibitor pretreatment were analyzed for levels of Ac-CoA by LC-MS. (D) BMDMs were stimulated or not for 12 hr with IL-4, followed by a 2 hr incubation with $^{13}C_6$-glucose, $^{13}C_{16}$-palmitate, or $^{13}C_5$-glutamine. Carbon tracing into Ac-CoA was assessed by LC-MS. Data shows arbitrary units of labeled $^{13}C$ (M+2) in the different conditions. The student's t-test was used to determine statistical significance, defined as $*P<0.05$, $**P<0.01$, and $***P<0.001$.

The following figure supplement is available for figure 3:

**Figure supplement 1.** Akt regulates Acly to control inducible Ac-CoA production in M2 macrophages.

## Akt regulates Acly phosphorylation to control Ac-CoA production in M2 macrophages

How might Akt regulate increased histone acetylation in M2 macrophages? We hypothesized that Akt may control production of Ac-CoA, the metabolic substrate for histone acetylation. Using quantitative stable isotope dilution-LC-MS, we found that IL-4 treatment led to a maximal increase in Ac-CoA levels of ~40–75% (*Figure 3A,C*). A key regulator of Ac-CoA production is the enzyme Acly, which cleaves cytosolic citrate to produce a nuclear-cytoplasmic pool of Ac-CoA (*Wellen et al., 2009*). Akt has been shown to phosphorylate and activate Acly (*Berwick et al., 2002*; *Lee et al., 2014*), and we found that in M2 macrophages, IL-4 treatment stimulated the activating phosphorylation of Acly in an Akt-dependent manner (*Figure 3B*, *Figure 3—figure supplement 1A*). Use of lysates from MEFs transfected with ACLY siRNA confirmed specificity in detection of phosphorylated and total Acly (*Figure 3—figure supplement 1B*). Importantly, cotreatment with Akt or Acly inhibitors blocked the IL-4-mediated increases in Ac-CoA levels (*Figure 3C*), indicating Akt- and Acly-mediated control of Ac-CoA production in M2 macrophages. Conversely, citrate, the substrate for the Acly reaction, accumulated in the presence of the inhibitors (*Figure 3—figure supplement 1C*).

Next, we asked about the carbon source of the Ac-CoA that supports optimal M2 gene induction. Untreated or IL-4-treated BMDMs were incubated with $^{13}C_6$-glucose, $^{13}C_{16}$-palmitate, $^{13}C_5$-glutamine, followed by carbon tracing into Ac-CoA as assessed by LC-MS (*Figure 3D*). IL-4 treatment enhanced $^{13}C$ (M+2) Ac-CoA labeling regardless of the tracer, indicating that all three metabolic fuels contributed to the elevated Ac-CoA pool. The highest labeling was observed in BMDMs fed palmitate. While LC-MS does not specifically measure the nuclear-cytosolic pool of Ac-CoA, these data suggests that palmitate may be the major carbon source for histone acetylation in M2 macrophages (*Figure 3D*).

## Acly regulates gene-specific histone acetylation to control M2 activation

These data prompted us to investigate a role for Acly in M2 activation. Indeed, the Acly inhibitor SB-204990 reduced IL-4-mediated induction of Akt-dependent M2 genes (*Arg1*, *Retnla*, *Mgl2*) but not Akt-independent M2 genes (*Chi3l3*, *Fabp4*, *Cd36*) (*Figure 4A*). The structurally distinct Acly inhibitor MEDICA 16 had similar effects, indicating specificity in inhibition (data not shown). Moreover, SB-204990 treatment attenuated IL-4-mediated increases in H3 and H4 acetylation at promoters of Akt-dependent M2 genes, but not Akt-independent M2 genes (*Figure 4B*, *Figure 4—figure*

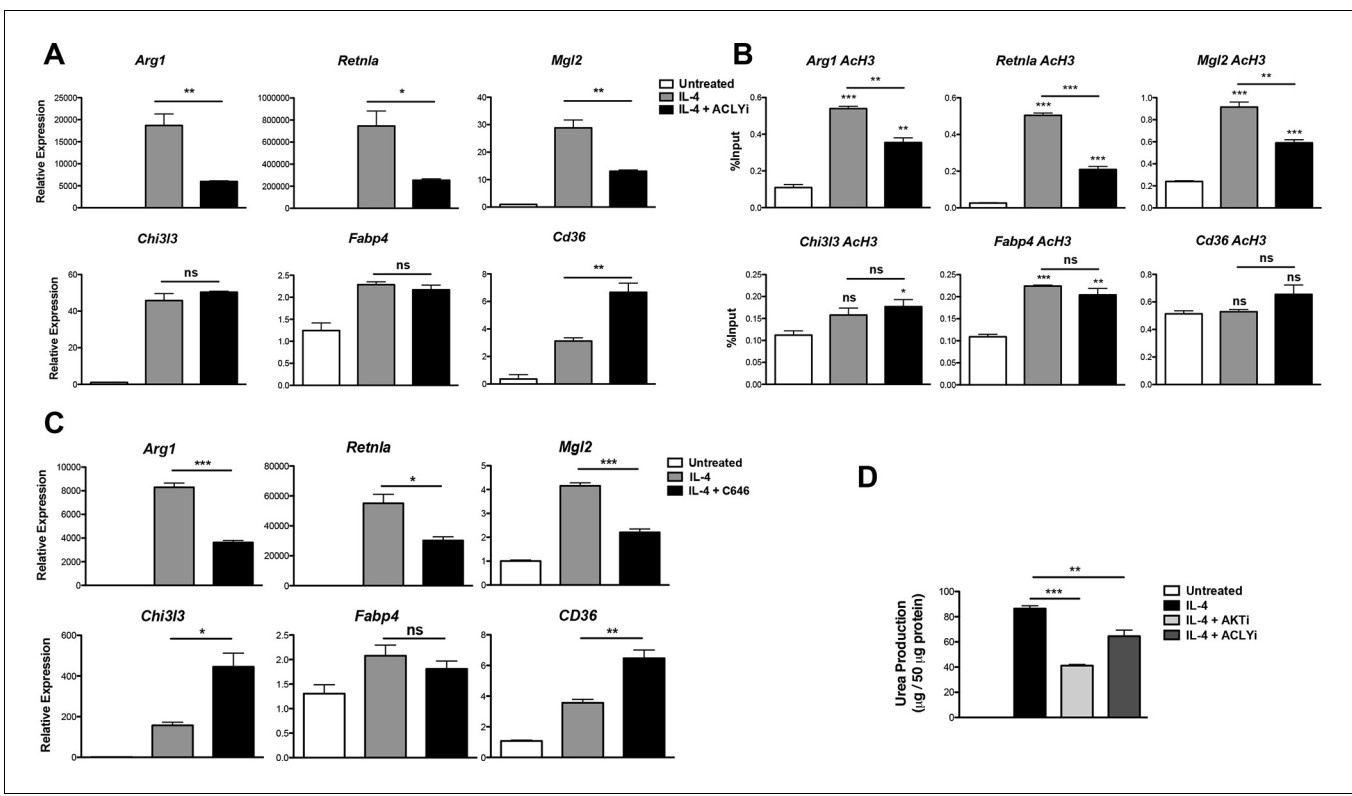

**Figure 4.** Acly controls inducible histone acetylation at some M2 genes. (A) Acly regulates induction of some M2 genes. BMDMs stimulated for 16 hr with IL-4 +/- Acly inhibitor pretreatment were analyzed for M2 gene induction by qRT-PCR. (B) Acly regulates inducible H3 acetylation at some M2 genes. BMDMs stimulated for 16 hr with IL-4 +/- Acly inhibitor pretreatment were subject to ChIP analysis using antibodies to acetylated H3. Enrichment of the indicated M2 gene promoters was assessed by qRT-PCR. (C) The p300 inhibitor C646 reduces induction of some M2 genes. BMDMs stimulated for 16 hr with IL-4 +/- C646 pretreatment were analyzed for M2 gene induction by qRT-PCR. (D) Akt and Acly control IL-4-inducible arginase activity. BMDMs were stimulated for IL-4 for 24 hr +/- inhibitor pretreatment, followed by analysis of arginase activity in cellular lysates as assessed by urea production. The student's t-test was used to determine statistical significance, defined as *$P<0.05$, **$P<0.01$, and ***$P<0.001$.

The following figure supplements are available for figure 4:

**Figure supplement 1.** Acly controls inducible histone acetylation at some M2 genes.

**Figure supplement 2.** Akt-Acly signaling regulates M2 activation in peritoneal macrophages.

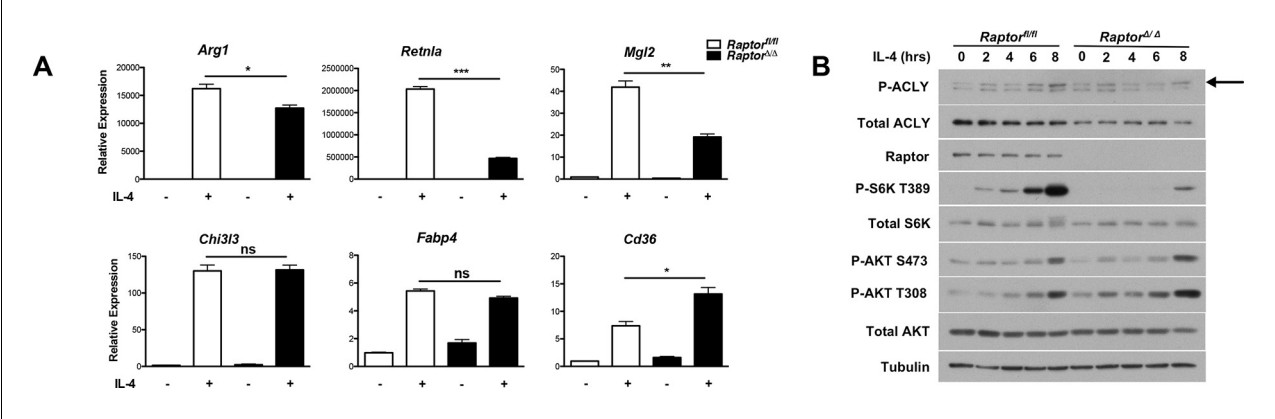

**Figure 5.** mTORC1 controls Acly protein levels to regulate M2 activation. (A) mTORC1 regulates M2 activation. M2 gene expression in *Raptor*fl/fl and *Raptor*△/△ BMDMs stimulated with IL-4 for 16 hr as assessed by qRT-PCR. (B) mTORC1 regulates Acly protein levels. Acly protein expression in *Raptor*fl/fl and *Raptor*△/△ BMDMs stimulated as indicated.

The following figure supplement is available for figure 5:

**Figure supplement 1.** mTORC1 activity regulates levels of Acly.

supplement 1A). Likewise, SB-204990 treatment diminished Pol II recruitment at Akt-dependent M2 genes (**Figure 4—figure supplement 1B**).

Because Akt and Acly regulate a global increase in Ac-CoA levels (**Figure 3C**) but control inducible histone acetylation only at some M2 gene promoters (**Figure 2D,E**, **4B**, and **Figure 4—figure supplement 1A**), Ac-CoA production is necessary but not sufficient for stimulating gene-specific increases in histone acetylation, which must be conferred by specific transcription factors and histone acetyltransferases (HATs). The activity of some HATs, including p300, is regulated by Ac-CoA levels and metabolic status (**Mariño et al., 2014**; **Pietrocola et al., 2015**). Interestingly, the p300 inhibitor C646 reduced induction of Akt-dependent but not Akt-independent M2 genes (**Figure 4C**). Therefore, p300 may link the Akt/Acly-dependent rise in Ac-CoA levels to increased histone acetylation and gene induction at some Akt-dependent M2 genes, while distinct HATs at Akt-independent genes are insensitive to such modulation of Ac-CoA levels.

Arginase activity is a hallmark feature of M2 activation that supports collagen production and polyamine synthesis (**Van Dyken and Locksley, 2013**). Consistent with effects on induction of *Arg1*, arginase activity was regulated by Acly and Akt (**Figure 4D**). Additionally, Akt and Acly inhibitors reduced induction of Akt-dependent M2 genes in peritoneal-elicited macrophages, indicating that control of M2 activation by the Akt-Acly axis may be applicable to multiple macrophage populations (**Figure 4—figure supplement 2**). Finally, induction of M2 gene expression by IL-13, a cytokine closely related to IL-4 that also triggers M2 activation (**Van Dyken and Locksley, 2013**), was also dependent on Akt and Acly (data not shown).

## mTORC1 regulates Acly protein levels in M2 macrophages

Our findings that Akt regulates Acly activity to control Ac-CoA production and M2 activation led us to consider a role for mTORC1 in this process. mTORC1 is a key downstream effector of Akt signaling and their activities are intricately linked in many settings ([**Dibble and Manning, 2013**; **Pollizzi and Powell, 2014**; **Laplante and Sabatini, 2012**] and **Figure 2—figure supplement 1B**). Indeed, we found that induction of Akt-dependent M2 genes was deficient in BMDMs lacking Raptor, a defining subunit of the mTORC1 complex (**Dibble and Manning, 2013**). In contrast, induction of Akt-independent M2 genes was not reduced (**Figure 5A**). mTORC1 is known to stimulate Acly expression (**Porstmann et al., 2008**; **Düvel et al., 2010**), and we found that *Raptor*-deficient BMDMs expressed lower levels of Acly protein (**Figure 5B**). Conversely, BMDMs with constitutive mTORC1 activity resulting from deletion of *Tsc1* (**Byles et al., 2013**), a negative regulator of mTORC1 (**Dibble and Manning, 2013**), displayed elevated Acly levels that were reduced by

treatment with the mTORC1 inhibitor rapamycin (*Figure 5—figure supplement 1*). Additionally, we noted that IL-4-inducible Acly phosphorylation was reduced in *Raptor*-deficient BMDMs (*Figure 5B*). This raises the possibility that mTORC1 could also regulate Acly activating phosphorylation, through mechanisms that remain to be clarified in future studies. Taken together, these data indicate that the Akt-mTORC1 axis controls Acly activating phosphorylation and protein levels, likely contributing to its control of M2 activation.

## The Akt-mTORC1 pathway couples metabolic input to induction of some M2 genes

The Akt-mTORC1 pathway is a major metabolic sensor, and mTORC1 activity in particular is controlled by amino acid levels, ADP/ATP levels, and other metabolic inputs (*Dibble and Manning, 2013*; *Laplante and Sabatini, 2012*). Therefore, we considered that incorporation of the Akt-mTORC1 pathway into IL-4 signaling, parallel to canonical Jak-Stat signaling, may allow particular subsets of the M2 transcriptional program to integrate signals reflecting the cellular metabolic state (*Figure 6A*). Amino acids directly and potently regulate mTORC1 activity independent of the TSC complex (*Dibble and Manning, 2013*; *Laplante and Sabatini, 2012*) and can also activate Akt in some contexts (*Tato et al., 2011*; *Novellasdemunt et al., 2013*), hence we varied amino acid concentrations as a way to modulate Akt-mTORC1 activity. As expected, mTORC1 activity, as assessed by phosphorylation of its downstream target S6K, was greatly reduced in amino acid deficient media and intermediate in media containing low levels of amino acids (*Figure 6B*). In line with (*Tato et al., 2011*; *Novellasdemunt et al., 2013*), increasing amino acid levels also augmented Akt activation, as indicated by enhanced phosphorylation on two critical residues, T308 and S473 (*Figure 6B*). Titrating amino acids had no effect on Stat6 phosphorylation and activation (*Figure 6B*), validating the use of this experimental model to modulate the Akt-mTORC1 axis independent of canonical Stat6 signaling. Consistent with effects on mTORC1 and Akt activity, amino acid levels dose dependently increased Acly phosphorylation and protein levels (*Figure 6B*) as well as Ac-CoA production (*Figure 6C*). Importantly, amino acids potentiated induction of Akt-dependent but not Akt-independent M2 genes (*Figure 6D*). This effect of amino acids was at least partially Raptor-dependent, indicating a critical role for mTORC1 in this process (*Figure 6—figure supplement 1*).

We also examined M2 activation using the complementary model of leucine deprivation, since leucine is particularly critical in regulation of mTORC1 activity (*Hara et al., 1998*). Here comparisons were made between culture conditions that differed only in the presence or absence of one amino acid, without significant effects on total levels of amino acids. Culture in leucine-deficient media attenuated IL-4-inducible mTORC1 and Akt activity and Acly phosphorylation, but not Stat6 phosphorylation (*Figure 6E*). Importantly, leucine deficiency selectively reduced expression of Akt-dependent M2 genes (*Figure 6F*). Taken together, these results indicate that amino acids and likely other metabolic inputs feed into the Akt-mTORC1 axis to calibrate M2 activation to the metabolic state (*Figure 6A*).

Finally, we found that physiological changes to nutrient levels can modulate M2 activation in adipose tissue macrophages (ATMs). ATM M2 polarization is thought to critically maintain insulin sensitivity in white adipose tissue, so such feeding-induced increases in M2 activation may coordinate responses to nutrient influx to mediate metabolic homeostasis in the postprandial state (*Odegaard and Chawla, 2011*). Specifically, we found that Akt activation was increased in the fed state compared to the fasted state in the ATM-containing stromal vascular (SVF) fraction of the white adipose tissue (*Figure 6—figure supplement 2A*). Although we were unable to reliably detect pAcly or Acly in the SVF for technical reasons, global H3 acetylation (*Figure 6—figure supplement 2A–B*) and M2 gene expression (*Figure 6—figure supplement 2C*) followed a similar pattern and were elevated in the fed state. Expression of all M2 genes was elevated in the fed state (*Figure 6—figure supplement 2C*), consistent with an important role for IL-13, a critical regulator of ATM M2 polarization (*Odegaard and Chawla, 2011*) that is increased in the fed state (*Figure 6—figure supplement 2D*), in feeding-induced ATM polarization, although postprandial elevations in nutrients like amino acids and glucose may also contribute. Therefore, feeding-inducible Akt activity correlated with increases in histone acetylation and M2 activation in ATMs.

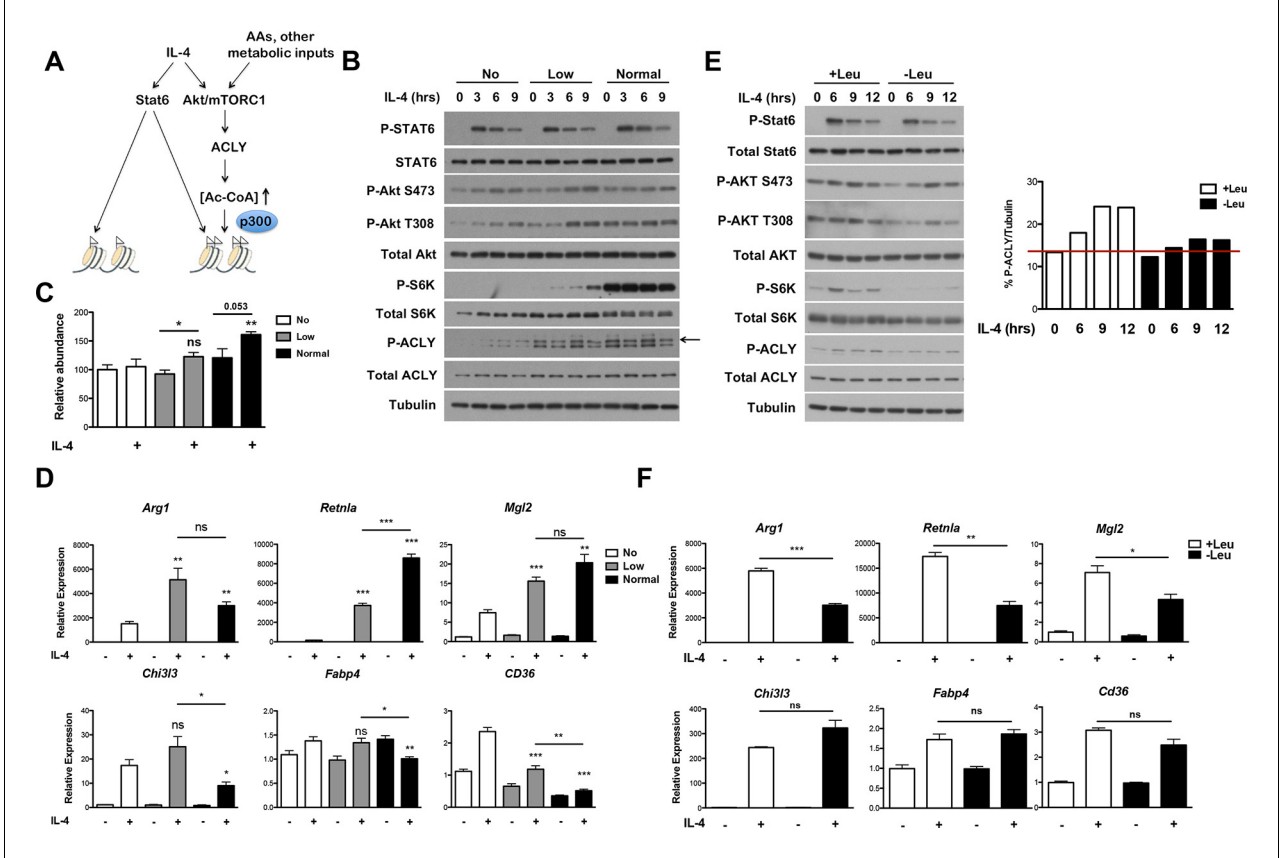

**Figure 6.** The Akt-mTORC1-Acly axis links metabolic input to control of M2 activation. (**A**) Proposed model for how Akt-mTORC1-Acly signaling exerts gene-specific control of M2 activation. Akt-TORC1-Acly signaling integrates metabolic input to control levels of Ac-CoA production, which modulates histone acetylation and gene induction at some M2 genes by HATs such as p300. (**B**) Amino acid levels modulate the activity of the Akt-mTORC1-Acly axis. BMDMs cultured in media containing varying levels of amino acids (normal, low, or no) were stimulated with IL-4 for the indicated time periods, followed by analysis of Akt, mTORC1, and Acly activity by western blotting. (**C**) Amino acid levels modulate Ac-CoA production. BMDMs stimulated as in B. were harvested for LC-MS analysis of Ac-CoA levels after 12 hr IL-4 stimulation. (**D**) Amino acid levels modulate induction of some M2 genes. BMDMs stimulated as in B. were harvested for qRT-PCR analysis of M2 gene induction after 9 hr IL-4 stimulation. (**E**) Leucine deficiency attenuates the activity of the Akt-mTORC1-Acly axis. BMDMs cultured in leucine-replete or leucine-deficient media were stimulated with IL-4 for the indicated time periods, followed by analysis of Akt, mTORC1, and Acly activity by western blotting. *Right,* quantitation of Acly phosphorylation. (**F**) Leucine deficiency reduces induction of some M2 genes. BMDMs stimulated as in E. were harvested for qRT-PCR analysis of M2 gene induction after 16 hr IL-4 stimulation. The student's t-test was used to determine statistical significance, defined as *$P<0.05$, **$P<0.01$, and ***$P<0.001$.

The following figure supplements are available for figure 6:

**Figure supplement 1.** Amino acid levels modulate M2 gene expression in part through Raptor.

**Figure supplement 2.** Feeding and fasting regulate M2 polarization of adipose tissue macrophages.

## Akt and Acly regulate functional subsets of the M2 program

We employed genome wide transcriptional profiling to obtain a comprehensive view of regulation of M2 activation by the Akt-Acly pathway. BMDMs were treated for 16 hr with IL-4 with or without Akt or Acly inhibitors, followed by RNA seq (*Figure 7*) or microarray analysis (data not shown). In the RNA seq analysis, 758 genes were induced >2.0 fold by IL-4, of which 91 were downregulated >30% by both Akt and Acly inhibitors (including *Arg1, Retnla*, and *Mgl2*), confirming critical roles for Akt and Acly in control of M2 activation as well as substantial overlap in the activities of the two proteins (*Figure 7A,B*). A subset of Akt inhibitor sensitive genes was sensitive to Acly inhibitor (91/327), in line with a broader role for Akt in control of cell physiology. In contrast, most genes sensitive to Acly inhibitor were sensitive to Akt inhibitor (91/118). This indicates that in the context of M2 activation,

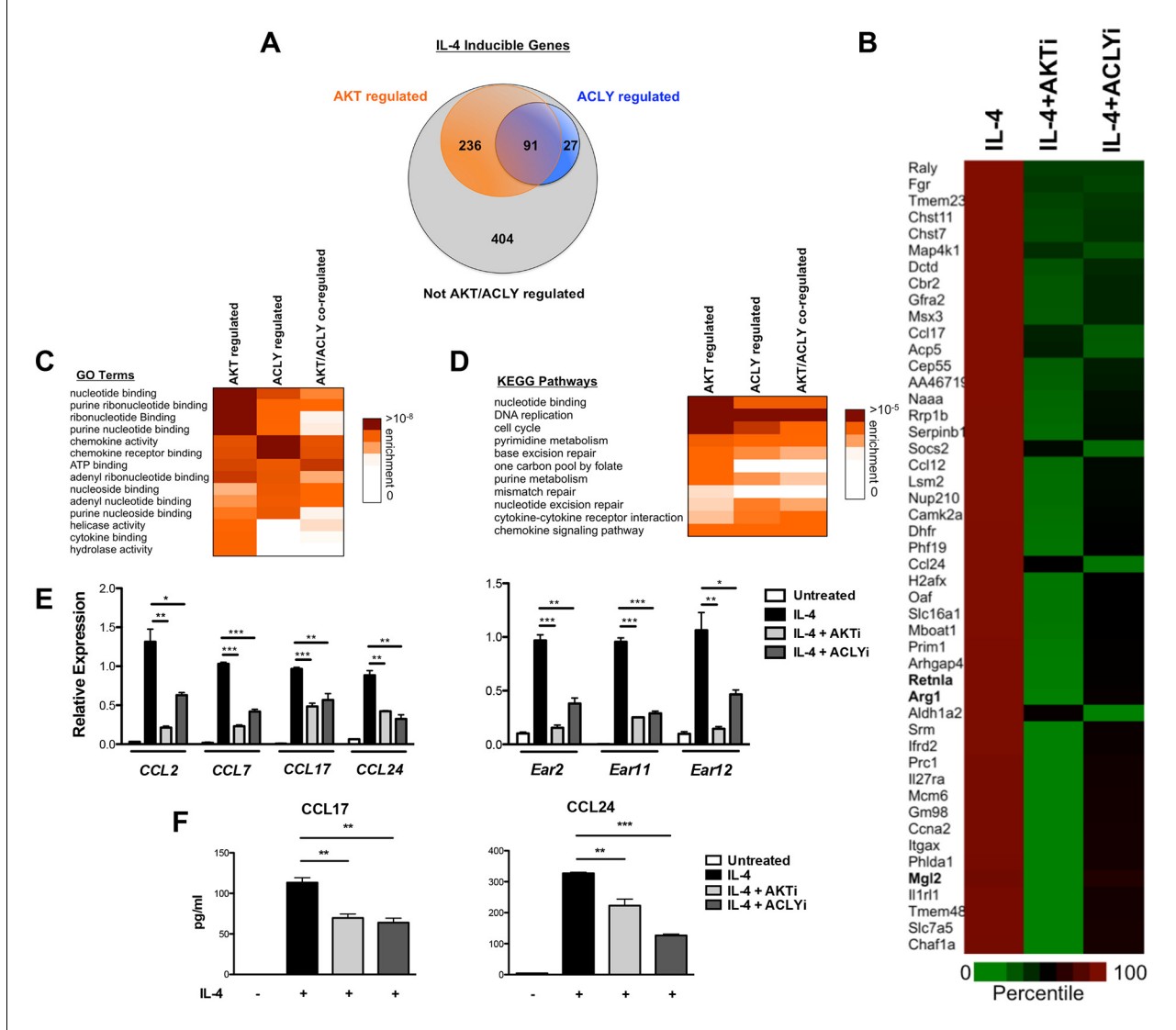

**Figure 7.** The Akt-Acly axis controls functional subsets of the M2 program. (**A**) Venn diagram depicting the number of IL-4-inducible genes regulated by Akt and/or Acly signaling. (**B**) Heatmap of normalized rank ordered Log2 RPKM values of top 50 IL-4 response genes co-regulated by Akt and Acly. (**C**) Heatmap of enriched KEGG pathways within the cohort of IL-4-inducible genes. (**D**) Heatmap of enriched Gene Ontology terms within the cohort of IL-4-inducible genes. (**E**) qPCR analysis validates regulation of chemokine and Ear genes by Akt-Acly signaling. BMDMs were stimulated with IL-4 for 16 hr +/- Akt or Acly inhibitor. (**F**) ELISA analysis indicates that Akt-Acly signaling regulates production of CCL17 and CCL24. BMDMs were stimulated with IL-4 for 36 hr +/- Akt or Acly inhibitor. The student's t-test was used to determine statistical significance, defined as *$P<0.05$, **$P<0.01$, and ***$P<0.001$.

The following figure supplement is available for figure 7:

**Figure supplement 1.** Working models.

Acly is a major target of Akt and is critically controlled by Akt activity, likely in regulation of Ac-CoA production and histone acetylation at M2 genes (**Figure 7A,B**).

Gene enrichment analysis of the 91 Akt- and Acly-coregulated genes identified preferential enrichment of several pathways, including cell cycle and DNA replication (**Figure 7C,D**). IL-4 triggered BrdU labeling of a subset of BMDMs in vitro (data not shown) and proliferation of macrophages in vivo (**Ruckerl et al., 2012**), thus IL-4 may stimulate macrophage proliferation in an Akt- and Acly-dependent manner. Consistently, metabolic processes underlying cellular proliferation were among the top enriched pathways in our metabolomics analysis, including nucleotide

metabolism and protein biosynthesis (*Figure 1A*). Interestingly, chemokines were also enriched in Akt- and Acly-coregulated genes (*Figure 7C–D*), including *Ccl2, Ccl7, Ccl17*, and *Ccl24*. Akt- and Acly-dependent induction of CCL17 and CCL24 was confirmed by qRT-PCR and ELISA (*Figure 7E–F*). Finally, genes in the eosinophil associated ribonucleases (Ear) family were found to be regulated by Akt and Acly. While barely missing the stringent cutoffs that we set for the RNA-seq analysis, qPCR analysis confirmed coregulation of *Ear2, Ear11*, and *Ear12* genes by the Akt-Acly pathway (*Figure 7E*). *Ear* genes are of interest because *Ear2* and *Ear11* are thought to have chemoattractant activity for dendritic cells and macrophages and are known to be highly induced in settings of Type 2 inflammation (*Cormier et al., 2002*; *Yamada et al., 2015*). Therefore, the transcriptional profiling analysis indicated that the Akt-Acly pathway controls selective subsets of the M2 program to allow their modulation by metabolic input (*Figure 7—figure supplement 1A*). As M2 macrophages play a key role in metabolic homeostasis, parasite infection, allergic diseases, and wound healing and tissue repair (*Van Dyken and Locksley, 2013*; *Odegaard and Chawla, 2011*), these findings are relevant for metabolic control of macrophage function in diverse contexts.

## Discussion

The Akt-mTORC1 pathway has a well-established role in promoting anabolic metabolism in growing/proliferating cells, tumor cells, and metabolic tissues. In the context of cellular proliferation, for example, Akt-mTORC1 activity couples growth factor signaling and nutrient availability to the synthesis of proteins, lipids, and nucleotides (*Dibble and Manning, 2013*). In contrast, the role of the Akt-mTORC1 pathway in macrophages is much less intuitive. What is the teleological rationale for control of macrophage activation by Akt-mTORC1 signaling (and metabolism more generally)?

Here we propose that IL-4 signaling *co-opted* the Akt-mTORC1 pathway to couple metabolic input to regulation of certain components of the M2 response, including chemokines and cellular proliferation (*Figure 7—figure supplement 1A*). This is supported by our findings that IL-4 signaling leads to parallel and independent activation of the Akt-mTORC1 pathway and the canonical Jak-Stat pathway, allowing the Akt-mTORC1 axis to regulate a subset of M2 genes through control of Acly activity/expression, Ac-CoA production, and histone acetylation. Why should some but not other components of the M2 response be regulated in this way? Control of cellular proliferation is intuitive, since Akt-mTORC1 signaling acts as a metabolic checkpoint in the context of cellular division to allow growth and proliferation only when nutrients are abundant. What about chemokines? We propose that chemokines may be controlled by the Akt-mTORC1 pathway because of their key role in amplifying energetically costly immune responses (*Hotamisligil and Erbay, 2008*). This allows metabolic status to calibrate immune responses such that inflammation is amplified and sustained only under metabolically favorable conditions. Interestingly, previous studies have shown that a critical role for Akt-mTORC1 signaling in activated CD8 T cells is to support their migration to sites of inflammation (*Finlay and Cantrell, 2011*). Therefore, Akt-mTORC1 signaling regulates both facets of immune response amplification, i.e., the ability of tissue-resident sentinel cells to mobilize activated leukocytes and of activated T cells to be recruited. Together these findings add another dimension to our emerging understanding of how metabolism supports leukocyte activation and immune responses.

As discussed above, the Akt/Acly-dependent rise in Ac-CoA production is necessary but not sufficient for stimulating gene-specific increases in histone acetylation. Such specificity is most likely conferred by HATs with distinct $K_m$ (*Pietrocola et al., 2015*). Indeed, our analysis suggests that p300 may preferentially regulate at least a subset of the Akt-dependent M2 genes (*Figure 4C*). Its high $K_m$ (*Mariño et al., 2014*; *Pietrocola et al., 2015*) may allow p300 to link metabolic status and Akt/mTORC1 activity, in the form of Ac-CoA levels, to histone acetylation and transcriptional induction at some M2 genes (*Figure 7—figure supplement 1B*). In contrast, HATs at Akt-independent M2 genes may have a low $K_m$ and are thus insensitive to such modulation of Ac-CoA levels. Presumably, differential HAT recruitment is mediated by distinct transcription factors at Akt-dependent and independent M2 genes, which would be important to address in future studies.

Although Akt activity has been linked to M2 activation, (*Byles et al., 2013*; *Ruckerl et al., 2012*), the role of mTORC1 remained unclear. Here, we use *Raptor△/△* BMDMs to show that mTORC1 activity stimulates M2 activation (*Figure 5A*). Furthermore, amino acids modulate mTORC1 activity (*Figure 6B*) to potentiate M2 gene induction in a Raptor-dependent manner (*Figure 6D*, *Figure 6—figure supplement 1*). Together these findings indicate that the Akt-mTORC1 signaling module

supports M2 activation. Acly appears to be a key target, with its expression levels and activating phosphorylation controlled by mTORC1 and Akt respectively. In seeming contrast to these data indicating that mTORC1 supports M2 activation, we and others have shown that aberrantly increased mTORC1 activity in *Tsc1*-deficient BMDMs attenuates M2 activation (*Byles et al., 2013*; *Zhu et al., 2014*). We hypothesize that the difference between the two models reflects divergent control of M2 activation by physiological and pathophysiological mTORC1 activity respectively. Downstream of the insulin receptor, such context-dependent roles of mTORC1 are well-established. In lean/healthy animals, mTORC1 critically mediates insulin signaling in metabolic tissues (to coordinate postprandial nutrient storage), but in obesity, chronic nutrient excess leads to an aberrant increase in mTORC1 activity that contributes directly to insulin resistance and metabolic dysregulation (*Laplante and Sabatini, 2012*). Similarly, while physiological mTORC1 activity couples metabolic input to M2 activation, pathophysiological mTORC1 activation during chronic nutrient excess may impair M2 activation. It would be interesting to see if the latter is true in adipose tissue macrophages in the context of diet-induced obesity, and if so, the consequences for tissue inflammation and metabolic homeostasis.

Interestingly, while inducible Akt phosphorylation occurred within minutes of IL-4 stimulation (*Figure 2B*), inducible Acly phosphorylation was detected with slightly delayed kinetics (~2 h, *Figure 5B* and data not shown). Such delay may reflect a need for other inputs that facilitate Akt-mediated Acly phosphorylation, or the reduced sensitivity and dynamic range of the pAcly antibody compared to the pAkt antibodies. Once pAcly is detectable at ~2 h, Akt and Acly phosphorylation nicely parallel and steadily increase up to (and perhaps beyond) 8 hr (*Figure 5B*). As expected, inducible Akt and Acly phosphorylation precede increases in global histone acetylation, which is observed starting only at 4 hr (*Figure 2C*). However, Ac-CoA levels increase only 8 hr after IL-4 stimulation (*Figure 3A*). One possibility, supported by the increase in global histone acetylation at 4 h, is that diversion of Ac-CoA into acetylated histones diminishes the free Ac-CoA pool. Another possibility is that because the LC-MS analysis measures bulk Ac-CoA rather than the nuclear-cytoplasmic pool relevant for histone acetylation, changes in mitochondrial Ac-CoA levels could be confounding. Again, global histone acetylation, which may more accurately reflect nuclear-cytoplasmic pools of Ac-CoA, increases 4 hr after IL-4 treatment (*Figure 2C*), as does gene-specific increases in histone acetylation at Akt-dependent M2 genes (*Figure 2—figure supplement 2C*). Therefore, we believe that the preponderance of the data support our model that IL-4 triggers Ac-CoA production and histone acetylation as a consequence of Akt-mediated Acly activation.

Metabolic status has long been proposed to modulate epigenetic control of gene expression (*Teperino et al., 2010*; *Kaelin and McKnight, 2013*; *Gut and Verdin, 2013*), but only recently have a handful of studies linked physiological changes in metabolite levels to chromatin regulation of gene expression (*Wellen et al., 2009*; *Lee et al., 2014*; *Shimazu et al., 2013*; *Carey et al., 2015*). Here we show how the Akt-mTORC1 axis couples metabolic input in the form of Ac-CoA levels to histone acetylation and gene regulation, and importantly, to control specific subsets of the M2 program. In addition to a recent study (*Lee et al., 2014*), this is only the second example of how Akt-Acly signaling controls gene regulation through histone acetylation. Other macrophage polarizing signals and common gamma chain cytokines (γc) (e.g. IL-2, IL-15) engage the Akt-mTORC1 axis, thus our findings may have implications for multiple programs of macrophage polarization and leukocyte activation. Canonical signaling downstream of the polarizing signal or γc specifies which genes are induced, while regulation of Ac-CoA levels and histone acetylation by the Akt-mTORC1-Acly pathway allows metabolic input to calibrate genes encoding energetically demanding processes; it would be informative in future studies to determine the nature of these processes. Alternatively, Ac-CoA can be synthesized independently of the Akt-mTORC1-Acly axis by AceCS1 (*Hallows et al., 2006*) or nuclear pyruvate dehydrogenase (*Sutendra et al., 2014*) to mediate histone acetylation. AceCS1 activity is controlled by SIRT1, thus providing a means for Ac-CoA production and histone acetylation in conditions of low energy or nutrients (*Hallows et al., 2006*). Therefore, future studies to determine how gene-specific histone acetylation is regulated during different macrophage activation programs are warranted. These studies could pave the way towards new therapeutic approaches of modulating macrophage function in diverse contexts, including Type 2 inflammation, metabolic homeostasis, and antimicrobial immunity.

## Materials and methods

### BMDM culture and stimulations

BMDM cultures were established as described (*Byles et al., 2013*). For stimulations, BMDMs were pretreated for 1 hr with inhibitors followed by addition of 10 ng/ml IL-4 for 16 hr unless otherwise indicated. Inhibitors were used as follows: AKT inhibitor MK-2206, 2–5 µM (Selleck, Houston, TX); ACLY inhibitor SB-204990, 40 µM (Tocris, United Kingdom); p300 inhibitor C646, 10 µM; etomoxir, 200 µM (Sigma, St. Louis, MO), and 2-deoxy-glucose, 1 mM (Sigma). For amino acid titration experiments, BMDMs were plated in DMEM containing low levels of amino acids for 6 hr (to deplete cellular amino acid pools) prior to changing the media to DMEM with varying levels of amino acids (no, low, or normal) +/- IL-4 for 16 hr. Normal is normal tissue culture media, while low indicates media containing 5% of the normal levels of amino acids (obtained by mixing normal media and media lacking amino acids). In experiments with leucine free media, BMDMs were stimulated in complete DMEM or –Leu complete DMEM (Crystalgen, Commack, NY) +/- IL-4 for 16 hr. *Tsc1*△/△ BMDMs were described previously (*Byles et al., 2013*). BMDMs from *UbiquitinC-CreERT2 Raptor*fl/fl mice were treated with tamoxifen to delete *Raptor*; parallel treatment of *Raptor*fl/fl BMDMs were used as controls.

### Mice

C57BL/6 mice were used for in vivo studies and as a source of BMDMs. Mice were maintained at Harvard Medical School and all procedures were performed in accordance with the guidelines set forth by the Institutional Animal Care and Use Committees at the institution. To generate *UbiquitinC-CreERT2 Raptor*fl/fl mice, previously described *Raptor*fl/fl mice (*Sengupta et al., 2010*) were crossed with UbiquitinC-CreERT2 mice (The Jackson Laboratory, Bar Harbor, ME) in David Sabatini's laboratory at the Whitehead Institute in Cambridge, Massachusetts, in accordance with the guidelines set forth by the Institutional Animal Care and Use Committee at the institution.

### Immunoblotting

Cells were lysed directly in 6X SDS loading buffer (histone western blots) or in 1% NP-40 buffer (all other western blots). Protein concentration was determined using the Bradford method. Primary antibodies were purchased from Cell Signaling except for α-Tubulin (Sigma), acetylated Tubulin (Sigma), acetylated H3 (Millipore, Germany), acetylated H4 (Millipore), and total H4 (Abcam, Cambridge, MA).

### Arginase assay

Arginase assay was done as described (*Byles et al., 2013*).

### Extracellular flux assays

Oxygen consumption and extracellular acidification rates were measured with a XF96 extracellular flux analyzer (Seahorse Bioscience, North Billerica, MA). Seahorse assay media containing 11 mM glucose or plain assay media was used for the mitochondrial and glycolysis stress tests respectively. OCR measurements were taken before and after the sequential addition of 1 µM oligomycin, 1.5 µM FCCP and 2 µM antimycin/rotenone (Sigma). ECAR measurements were taken before and after the sequential addition of 11 mM glucose, 1 µM oligomycin and 0.5 M 2-DG (Sigma). Values were normalized with Hoechst 33342 staining (Life Technologies, Carlsbad, CA).

### Glucose uptake

BMDMs were washed with Krebs-Ringer bicarbonate HEPES (KRBH) buffer once, followed by addition of 400 µl KRBH buffer. 100 µl loading buffer (KRBH buffer with 0.5 mM 2-deoxy-D-glucose (Sigma) and 1 µCi/well $^3$H-deoxy-D-glucose (2-$^3$H[G]) (PerkinElmer, Waltham, MA, 1 mCi/ml in EtOH: water [9:1]) was added and incubated at 37°C for exactly 15 min. 20 µl stop solution (1.5 mM cytochalasin B (Sigma) in DMSO) was added, and the cells were washed with KRBH buffer before lysis in 0.1 N NaOH. The glucose uptake rate was determined by normalizing cellular $^3$H-deoxy-D-glucose count to protein concentrations.

## Fatty acid oxidation

Fatty acid oxidation was done as described (*Byles et al., 2013*).

## Chromatin immunoprecipitation

ChIP was done as described (*Byles et al., 2013*), using acetylated H3 (Millipore 06–599), acetylated H4 (Millipore 06–866), or IgG (Santa Cruz, Dallas, TX, SC-2027) antibodies. Fold enrichment was calculated as ChIP signals normalized to input. ChIP primer sequences as well as position relative to transcription start site (TSS) are provided in *Supplementary file 2*.

## Gene expression

RNA was isolated using RNA-Bee (Tel-Test, Friendswood, TX) per manufacturers protocol. cDNA synthesis was done using High Capacity cDNA Reverse Transcription Kit (Applied Biosystems, Foster City, CA). A Bio-Rad C1000 Thermocycler was used for qPCR, and data was analyzed by means of the CFX Manger Software (Bio-Rad, Hercules, CA) using the delta/delta CT method. BMDM samples were normalized to hypoxanthine phosphoribosyltransferase while ex vivo samples were normalized to the macrophage marker CD68.

## Dual luciferase assays

BMDMs were electroporated using mouse macrophage nucleofector kit (Lonza, Hopkinton, MA) and the Amaxa machine with STAT6-Firefly luciferase (Addgene, Cambridge, MA, plasmid #35554) along with Renilla–Luciferase plasmid as a transfection control. BMDMs were stimulated with or without 10 ng/ml IL-4 4 hr post electroporation for another 24 hr. Cell lysates were collected and analyzed using the Promega Dual-Luciferase Reporter Assay System.

## Acyl-CoA mass spectometry

BMDMs were lysed in 800 µl ice cold 10% TCA (Tricholoracetic acid). Sc5-sulfosalicylic acid (SSA), ammonium formate, [$^{13}C_6$]-glucose, sodium [$^{13}C_{16}$]-palmitate, and analytical standards for acyl-CoAs were from Sigma-Aldrich (St. Louis, MO). Optima LC-MS grade methanol, ammonium acetate, acetonitrile (ACN) and water were purchased from Fisher Scientific (Pittsburgh, PA). Calcium [$^{13}C_3{}^{15}N_1$]-pantothenate was purchased from Isosciences (King of Prussia, PA). [$^{13}C_3{}^{15}N_1$]-acyl-CoA internal standards for quantitation were generated by pan6 deficient yeast culture as previously described (*Snyder et al., 2015*), with 100 µL of extract spiked into samples before extraction. Standard curves were prepared using the same batch of internal standard, and all samples were extracted by solid phase extraction as previously described (*Basu and Blair, 2012*). Acyl-CoAs were analyzed as previously described for quantitation (*Basu et al., 2011*) and for isotopolog analysis (*Worth et al., 2014*) by liquid chromatography-tandem mass spectrometry on an Agilent 1200 coupled to an API4000 in the positive ion mode monitoring the acyl-CoA specific neutral loss of 507 amu from each acyl-CoA, internal standard and isotopolog. For carbon tracing experiments, BMDMs were treated with 10 ng/ml IL-4 for 12 hr before the addition of tracers (2g/L $^{13}C_6$-glucose, 50 µM $^{13}C_{16}$-palmitate, or 2 mM $^{13}C_5$-glutamine) for another 2 hr.

## Steady state metabolomics

BMDMs were stimulated for 10 hr with IL-4 before media was refreshed by addition of complete RPMI with IL-4 for another 2 hr. Preparation of cellular extracts was done as described (*Ben-Sahra et al., 2013*). Steady state metabolomics was done at Beth Israel Deaconess Medical Center Mass Spectrometry Facility. Data analysis was performed as described (*Ben-Sahra et al., 2013*).

## RNA-seq library construction, mapping, and analysis

Strand-specific libraries were generated using 500ng RNA input using TruSeq library preparation kit (Illumina, San Diego, CA). cDNA libraries were multiplexed using specific unique adaptors and sequenced using Illumina NextSeq 500 under single end 75bp read length parameters. Reads were aligned to the mouse mm10 reference genome using TopHat using default settings (*Langmead et al., 2009*). Alignments were restricted to uniquely mapping reads, with up to 2 mismatches permitted. RPKM was calculated as described for mm10 Refseq genes by counting exonic reads and dividing by mRNA length (*Mortazavi et al., 2008*). Coexpressed gene classes were generated with Cluster3 by

applying k-means clustering to mean-centered log2 (FPKM) expression values. Differential analyses was performed using DEseq (*Anders and Huber, 2010*) using default parameters for the indicated comparisons. Cohort of IL-4 inducible genes was defined by following: >2 RPKM, Log2fold>1.0, DESeq *P*-adj<0.05 yielding 758 IL-4 inducible genes. Inhibition by AKT or ACLY inhibitors defined as 30% reduction in RPKM and DESeq *P*-adj <0.05. Enrichment of KEGG pathways and Gene Ontology (GO) terms analysis performed using DAVID (*Huang et al., 2008*).

## Feeding/fasting experiments

8–10 week old C57BL/6 mice were fasted overnight or allowed to feed ad-libitum. Mice were sacrificed the next morning and the perigonadal adipose tissue was excised. A small section of whole adipose tissue (WAT) was homogenized in RNA-Bee for analysis of gene expression in unfractionated WAT. The remaining adipose tissue was minced and digested in 5 ml Krebs ringer buffer (KRBH) containing 2% fatty acid free BSA and 2 mg/ml collagenase (Sigma, C2674) for 20 min at 37°C. The resulting cell suspension was filtered through a 250 mm nylon mesh and centrifuged at 1200 RPM to obtain a cell pellet corresponding to the stromal vascular fraction (SVF), which was lysed for RNA extraction or western blotting.

## Statistical analysis

Statistical analysis was carried out using Prism (GraphPad) software. The student's t-test was used to determine statistical significance, defined as *$P<0.05$, **$P<0.01$, and ***$P<0.001$.

## Acknowledgements

This project was supported by a NIH grant R01AI102964 (to TH), CURE grant (to NWS), and R35CA197459 (to BDM). AJC is a recipient of the Ford Foundation Predoctoral Fellowship and the Ford Foundation Dissertation Fellowship. DL was supported by a K99/R00 Pathway to Independence Award from the NIH/NIA (AG041765). We thank CH Lee for providing Stat6 KO mice and DM Sabatini for providing bone marrow from *Raptor*[fl/fl] and *UbC-CreERT2 Raptor*[fl/fl] mice. All authors have reviewed the manuscript and declare no competing interests.

## Additional information

### Funding

| Funder | Grant reference number | Author |
|---|---|---|
| National Institutes of Health | R01AI102964 | Tiffany Horng |
| Ford Foundation | Predoctoral Fellowship | Anthony J Covarrubias |
| National Institutes of Health | R35CA197459 | Brendan D Manning |
| Commonwealth Universal Research Enhancement | Pennsylvania Department of Health grant | Nathaniel W Snyder |
| National Institutes of Health | AG041765 | Dudley Laming |

The funders had no role in study design, data collection and interpretation, or the decision to submit the work for publication.

### Author contributions

AJC, HIA, JY, NWS, AJW, VB, TPS, ECE, Acquisition of data, Analysis and interpretation of data, Drafting or revising the article; SSI, JW, IBS, BDM, YZ, IAB, Analysis and interpretation of data, Drafting or revising the article; DL, Acquisition of data, Drafting or revising the article, Contributed unpublished essential data or reagents; TH, Conception and design, Analysis and interpretation of data, Drafting or revising the article

### Author ORCIDs

Halil Ibrahim Aksoylar, http://orcid.org/0000-0002-0527-6124

## Ethics

Animal experimentation: Mice were maintained at Harvard Medical School and all procedures were performed in accordance with approved Institutional Animal Care and Use Committee protocol #04549 at the institution.

## Additional files

### Supplementary files

• Supplementary file 1. LC-MS peak areas and *P*-values for experiment described in *Figure 1A* (ranked by -*P*-values of pairwise comparison).

• Supplementary file 2. Sequences of ChIP primers, as well as their positions relative to the transcriptional start site (TSS).

### Major datasets

The following datasets were generated:

| Author(s) | Year | Dataset title | Dataset URL | Database, license, and accessibility information |
|---|---|---|---|---|
| Anthony J Covarrubias, Halil Ibrahim Aksoylar, Jiujiu Yu, Nathaniel W Snyder, Andrew J Worth, Shankar S Iyer, Jiawei Wang, Issam Ben-Sahra, Vanessa Byles, Tiffany Polynne-Stapornkul, Erika C Espinosa, Dudley Laming, Brendan D Manning, Yijing Zhang, Ian A Blair, Tiffany Horng | 2015 | Akt-Acly signaling integrates metabolic input to epigenetic control of macrophage activation | http://dx.doi.org/10.5061/dryad.3h061 | Available at Dryad Digital Repository under a CC0 Public Domain Dedication |

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
