## [Decision Letter]

Thank you for submitting your work entitled "Akt-Acly signaling integrates metabolic input to epigenetic control of macrophage activation" for consideration by *eLife*. Your article has been reviewed by two peer reviewers, and the evaluation has been overseen by a Reviewing Editor and Tadatsugu Taniguchi as the Senior Editor.

The reviewers have discussed the reviews with one another and the Reviewing editor has drafted this decision to help you prepare a revised submission.

Summary:

This study shows that Act stimulation in macrophages by IL-4, a cytokine that dictates alternative (M2) activation of these cells, leads to activation of Acly, an enzyme that mediates Ac-CoA production, and thus enhances histone acetylation and induction of a subset of the M2 genes. It further shows that this Act effect is subject to modulation by metabolic signals, which probably serves to adjust the differentiation state and function of macrophages to nutrient availability.

Essential revisions:

1) A major concern was raised on the timing by which various reported events occur and the implications of this for the overarching model. AKT phosphorylation increases within 15 min of IL-4 stimulation, histone acetylation increases by about 4 hours, while pACLY increases by 6 hours, and acetyl-CoA by 8 hours. This does not seem to be consistent with the proposed model, in which pACLY is directly downstream of AKT and should result in increased acetyl-CoA production proceeding or concurrent with increased histone acetylation. It is suggested that all of these parameters (or at least pAKT, pACLY, and histone acetylation) should be examined in the same experiment over the same time course to clarify whether the proposed model truly makes sense.

2) The contribution of mTORC1 here should be further clarified. Based on the data, it doesn't look like mTORC1 is a major contributor to the IL-4-driven pathway, since mTORC1 is implicated in regulating ACLY protein levels and ACLY protein levels do not increase with IL-4. Yet, mTORC1 is included as a major component of the paper's conclusions. Would authors agree that the data don't appear to support mTORC1 as a major physiological component of the mechanism?

---

## [Author Response]

Essential revisions:

*1) A major concern was raised on the timing by which various reported events occur and the implications of this for the overarching model. AKT phosphorylation increases within 15 min of IL-4 stimulation, histone acetylation increases by about 4 hours, while pACLY increases by 6 hours, and acetyl-CoA by 8 hours. This does not seem to be consistent with the proposed model, in which pACLY is directly downstream of AKT and should result in increased acetyl-CoA production proceeding or concurrent with increased histone acetylation. It is suggested that all of these parameters (or at least pAKT, pACLY, and histone acetylation) should be examined in the same experiment over the same time course to clarify whether the proposed model truly makes sense.*

We thank the reviewers for these suggestions and have redone the pAKT, pACLY, and histone acetylation blots using the same time courses. Inducible pAKT is detected very rapidly (Figure 2, Figure 5) while inducible pACLY is detected starting at ~2 h (Figure 5 and data not shown). The lag in ACLY activation relative to AKT activation may reflect a need for other inputs that facilitate AKT-mediated ACLY phosphorylation; alternatively, the large dynamic range of AKT phosphorylation (relative to ACLY phosphorylation), likely reflecting the extreme sensitivities of the pAKT antibodies, may enable earlier detection of pAKT. In any case, once pACLY is detectable at ~2h, AKT and ACLY phosphorylation nicely parallel and steadily increase up to (and perhaps beyond) 8 h (Figure 5). The kinetics of Akt and Acly phosphorylation precede that of global histone acetylation, which is detectable starting at 4 h but not at 2 h (Figure 2). These data support our model that IL-4 triggers Akt-mediated Acly activation and consequent histone acetylation. (Note that cells are lysed directly in 6x SDS buffer for the histone acetylation blots but NP-40 lysis buffer for the Akt and Acly blots so these are not the same experiments, but we have used the same time course to facilitate direct comparison of the two sets of data. Also, both the Akt/Acly activation and the histone acetylation experiments have been done multiple times over similar time courses with reproducible results, so we believe that experiment to experiment variability is unlikely.)

Not consistent with the Akt and Acly activation and histone acetylation, Ac-CoA levels increase at 8 h but not 4 h (Figure 3). One possibility, supported by the increase in global histone acetylation at 4 h, is that diversion of Ac-CoA into acetylated histones lowers the Akt- and Acly-dependent increase in the free Ac-CoA pool. Another possibility is that because our LC-MS analysis measures bulk Ac-CoA rather than the nuclear-cytoplasmic pool that is relevant for histone acetylation, reduced mitochondrial Ac-CoA may offset an early elevation in the nuclear-cytosolic pool of Ac-CoA. Unfortunately, current technologies for Ac-CoA measurements are incompatible with subcellular fractionation, precluding a direct test of this hypothesis. In this regard, the global histone acetylation may be an important complementary analysis that more accurately reflects nuclear-cytoplasmic pools of Ac-CoA. As noted above, the global histone acetylation increases ~4 h after IL-4 stimulation (Figure 2), with similar kinetics to gene-specific increases in histone acetylation (Figure 2—figure supplement 2). Taken together, we believe that the preponderance of the data including a careful kinetic analysis of pAkt, pAcly, and histone acetylation support our working model. An abbreviated version of this discussion is also included in the manuscript.

At this point we would also like to address the reliability of the histone acetylation blots. First, we would like to clarify that the histone acetylation blots are normalized to tubulin and are evenly loaded based on total protein (Figure 2, Figure 2—figure supplement 2). The H3 and H4 acetylation are then normalized to total H3 and H4 respectively, and then quantified by Image J densitometry. Using this protocol, the IL-4-inducible increase in global histone acetylation (normalized for the increase in total histones) is modest but very consistent and reproducible in several independent experiments (representatives of which are shown in the manuscript). We also tried acid extraction of histones as suggested by the reviewer (Epigentek, EpiQuik Total Histone Extraction Kit, Cat # OP-0006). For reasons not clear to us, multiple attempts with the acid extraction protocol indicate no IL-4-inducible increase in global histone acetylation (measured at 2 h intervals for up to 12 h following IL-4 stimulation; data not shown). We are inclined to believe the data obtained with the original protocol rather than the acid extraction for the following reasons: 1) The original protocol yielded extremely consistent/reproducible results over several independent experiments; 2) The correlation of increased histone acetylation with cellular proliferation seen in the original protocol is observed in many other contexts (e.g. PMID: 21596309); 3) Using the original protocol, addition of acetate or the HDAC inhibitor TSA led to the expected increases in global H3 and H4 acetylation (data not shown); and 4) That IL-4-inducible total histone acetylation is Akt-dependent (Figure 2—figure supplement 2) and starts ~4 h is consistent with the other data that we have presented in the manuscript. Therefore, we have left the histone acetylation blots in the manuscript, but defer to the judgment of the reviewers and can remove these blots if needed. A final consideration for keeping the histone acetylation blots in the manuscript is that, as noted above, global histone acetylation may more accurately reflect nuclear-cytoplasmic pools of Ac-CoA than LC-MS measurements of bulk Ac-CoA.

*2) The contribution of mTORC1 here should be further clarified. Based on the data, it doesn't look like mTORC1 is a major contributor to the IL-4-driven pathway, since mTORC1 is implicated in regulating ACLY protein levels and ACLY protein levels do not increase with IL-4. Yet, mTORC1 is included as a major component of the paper's conclusions. Would authors agree that the data don't appear to support mTORC1 as a major physiological component of the mechanism?*

We thank the reviewers for pointing out an opportunity for strengthening our manuscript. We believe that in addition to and together with Akt, mTORC1 regulates a subset of M2 genes and critically couples metabolic input to induction of these genes. We have now obtained Raptor-deficient BMDMs, and showed that loss of mTORC1 leads to reduced expression of the Akt-dependent M2 genes *Arg1, Fizz1, Mgl2, but not Ym1, Fabp4,* and *Cd36* (Figure 5). This provides strong genetic evidence that physiological mTORC1 activity supports M2 activation. Moreover, the ability of amino acids to modulate Akt-dependent M2 genes is attenuated in Raptor-deficient BMDMs, indicating that mTORC1 is an important part of the mechanism by which amino acid availability calibrates this subset of M2 genes (Figure 6—figure supplement 1). Taken together, these findings implicate physiological mTORC1 activity in control of M2 activation and in linking metabolic input to a subset of M2 genes.

We believe that an important role of mTORC1 in its control of M2 activation is regulation of Acly protein levels. Raptor KO BMDMs lacking mTORC1 have clearly reduced levels of Acly protein (Figure 5). Conversely, TSC KO BMDMs with constitutively elevated mTORC1 activity have elevated ACLY levels that are sensitive to inhibition with rapamycin (Figure 5—figure supplement 1). Moreover, amino acids dose dependently augment total ACLY levels (Figure 6), most likely through mTORC1. Although IL-4 only stimulates a very modest to negligible increase in Acly protein levels (as the reviewers point out), we believe that serum may be driving mTORC1-stimulated Acly protein levels at steady state thus confounding the ability to detect IL-4-inducible increases. In this regard it is interesting to note the clear reduction in the IL-4-inducible pAcly in Raptor-deficient BMDMs (Figure 5). While the underlying mechanism remains to be clarified in future studies, this raises the possibility that in addition to Akt, mTORC1 could regulate Acly phosphorylation. Taken together, these findings strongly indicate an important role for mTORC1 in controlling Acly protein levels (and perhaps also phosphorylation) in M2 macrophages.

Taken together, we believe that our data indicates that mTORC1 couples metabolic input to M2 activation through Acly (and perhaps other mechanisms). We have elaborated on this in the revised manuscript.